# Root Cause Analysis to Identify Medication and Non-Medication Strategies to Prevent Infection-Related Hospitalizations from Australian Residential Aged Care Services

**DOI:** 10.3390/ijerph17093282

**Published:** 2020-05-08

**Authors:** Janet K. Sluggett, Samanta Lalic, Sarah M. Hosking, Brett Ritchie, Jennifer McLoughlin, Terry Shortt, Leonie Robson, Tina Cooper, Kelly A. Cairns, Jenni Ilomäki, Renuka Visvanathan, J. Simon Bell

**Affiliations:** 1Centre for Medicine Use and Safety, Faculty of Pharmacy and Pharmaceutical Sciences, Monash University, Parkville, VIC 3052, Australia; samanta.lalic@monash.edu (S.L.); s.hosking@deakin.edu.au (S.M.H.); jenni.ilomaki@monash.edu (J.I.); simon.bell2@monash.edu (J.S.B.); 2University of South Australia, Adelaide 5001, Australia; 3NHMRC Cognitive Decline Partnership Centre, Hornsby Ku-ring-gai Hospital, Hornsby 2077, Australia; 4Pharmacy Department, Monash Health, Melbourne 3168, Australia; 5National Health and Medical Research Council of Australia Centre of Research Excellence in Frailty and Healthy Aging, Adelaide 5005, Australia; renuka.visvanathan@adelaide.edu.au; 6Infectious Diseases Department, Women’s and Children’s Hospital, Adelaide 5006, Australia; Brett.Ritchie@sa.gov.au; 7Resthaven Incorporated, Adelaide 5034, Australia; jmcloughlin@resthaven.asn.au (J.M.); TShortt@resthaven.asn.au (T.S.); LRobson@resthaven.asn.au (L.R.); TCooper@resthaven.asn.au (T.C.); 8Pharmacy Department, The Alfred, Alfred Health, Melbourne, VIC 3181, Australia; k.cairns@alfred.org.au; 9Department of Epidemiology and Preventive Medicine, Monash University, Melbourne, VIC 3004, Australia; 10School of Medicine, Faculty of Health and Medical Sciences, University of Adelaide, Adelaide 5005, Australia; 11Aged and Extended Care Services, The Queen Elizabeth Hospital, Central Adelaide Local Health Network, SA Health, Adelaide 5011, Australia

**Keywords:** infection, residential aged care, long-term care, hospitalization, root cause analysis, antimicrobial stewardship, medication review, Australia

## Abstract

Infections are leading causes of hospitalizations from residential aged care services (RACS), which provide supported accommodation for people with care needs that can no longer be met at home. Preventing infections and early and effective management are important to avoid unnecessary hospital transfers, particularly in the Australian setting where new quality standards require RACS to minimize infection-related risks. The objective of this study was to examine root causes of infection-related hospitalizations from RACS and identify strategies to limit infections and avoid unnecessary hospitalizations. An aggregate root cause analysis (RCA) was undertaken using a structured local framework. A clinical nurse auditor and clinical pharmacist undertook a comprehensive review of 49 consecutive infection-related hospitalizations from 6 RACS. Data were collected from nursing progress notes, medical records, medication charts, hospital summaries, and incident reports using a purpose-built collection tool. The research team then utilized a structured classification system to guide the identification of root causes of hospital transfers. A multidisciplinary clinical panel assessed the root causes and formulated strategies to limit infections and hospitalizations. Overall, 59.2% of hospitalizations were for respiratory, 28.6% for urinary, and 10.2% for skin infections. Potential root causes of infections included medications that may increase infection risk and resident vaccination status. Potential contributors to hospital transfers included possible suboptimal selection of empirical antimicrobial therapy, inability of RACS staff to establish on-site intravenous access for antimicrobial administration, and the need to access subsidized medical services not provided in the RACS (e.g., radiology and pathology). Strategies identified by the panel included medication review, targeted bundles of care, additional antimicrobial stewardship initiatives, earlier identification of infection, and models of care that facilitate timely access to medical services. The RCA and clinical panel findings provide a roadmap to assist targeting services to prevent infection and limit unnecessary hospital transfers from RACS.

## 1. Introduction

Residents of aged care services often live in close proximity to one another, have comorbid conditions, and have unavoidable contact with health care workers. These conditions are conducive to rapid infection transmission and increase the risk of morbidity and mortality from infectious diseases [1]. Infections are one of the leading causes of hospitalization from residential aged care services (RACS) [1,2,3,4,5]. RACS are synonymous with “nursing homes” and “long-term care facilities” and provide supported accommodation for people with care needs that can no longer be met in their own homes [4,6]. Australian and European studies have found that up to 25% of all hospitalizations from RACS are for infection [3,4], most commonly for respiratory, urinary tract, gastrointestinal, and skin infections [2,5,7,8]. One study in the United States (US) found that potentially preventable hospitalizations accounted for 23% (USD 223.8 million) of the total cost of hospitalizations from RACS in 2004 [9]. Heterogeneity in RACS settings and different definitions of “preventable” means that the proportion of hospitalizations deemed potentially preventable varies [2]. However, previous research suggests that 13%–67% of infection-related hospitalizations are potentially preventable [2,5,10], and therefore preventing unnecessary hospitalizations is a priority for RACS providers.

Broad strategies for preventing infection-related hospitalization may aim to prevent an infection occurring (e.g., vaccinations) or to better manage an infection in the RACS to avoid hospitalization (e.g., early detection and administering appropriate antimicrobials). Prevention of infection in RACS is important as there is increasing concern regarding antimicrobial resistance [11]. Antimicrobial resistance is associated with increased hospital costs and length of stay and death [12]. Antimicrobial stewardship, outbreak control and initiatives to prevent urinary tract infections (UTIs) have strengthened in the US RACS from 2013 to 2018 [13]. A recent systematic review found high-quality evidence to suggest that vaccinating residents against influenza reduces hospitalizations from RACS [14]. Strategies to prevent infections include general infection control procedures, such as promoting hand hygiene [8,15]. Other strategies include ongoing staff education in infection control [8], effective communication between staff and with external healthcare providers [8], environmental cleaning, and use of personal protective equipment such as gloves and gowns [16]. These are the same key principles of infection prevention and control outlined by the Australian Government for RACS providers [17]. Hospitalizations for infectious diseases may be reduced if advance care directives are put in place on admission to RACS and reviewed when a resident’s condition changes and/or deterioration in resident condition suggestive of infection is identified earlier [5]. Hospitalizations may also be reduced with effective communication among staff, and/or the management of infection at the RACS with the resources available, or with new models of care that facilitate provision of medical services that are not routinely available in Australian RACS [2,10,18].

Several strategies have been developed and trialed to prevent specific types of infection in RACS [16], although evidence regarding effectiveness of these strategies has been mixed [19]. These include protocols to reduce the number of catheter-associated UTIs [20], clinical care standards on infection [21], toolkits and protocols for preventing and managing gastroenteritis outbreaks [22], and protocols for effective monitoring and care of wounds including diabetic ulcers, pressure injuries, surgical wounds, and other injuries [16]. Adequate oral care for residents [23], identification of dysphagia and aspiration risk protocols [24], and pneumococcal [25] and influenza vaccination among residents [14,25] and RACS staff [26] have been recommended for prevention of respiratory infections.

A root cause analysis (RCA) is a process undertaken in healthcare settings to understand the underlying factors that led to a specific event of interest and develop strategies to help avoid similar occurrences in the future [27,28]. Previous research from the US has shown that an aggregate RCA process, which investigates a group of similar events, can be used to identify and develop strategies to prevent hospitalizations from skilled nursing facilities [18]. However, this strategy has yet to be applied in an Australian setting in the investigation of infection-related hospitalizations specifically.

In Australia, the formal, subsidized interdisciplinary antimicrobial stewardship programs that exist within the hospital setting are not routinely available in Australian RACS. However, new national Aged Care Quality Standards that apply from July 2019 outline the need for RACS provider organizations to implement antimicrobial stewardship policies and activities [29]. One quarter of all hospitalizations from South Australian RACS are for infections [4]. An improved understanding of strategies that could be applied to reduce infection risk and hospital transfers locally could assist stakeholders to enhance resident quality of care. The objective of this study was to examine root causes of infection-related hospitalizations from RACS and identify strategies to limit infections and avoid unnecessary hospitalizations among residents of aged care services. 

## 2. Materials and Methods

### 2.1. Design and Setting

This study was an aggregate RCA that utilized the South Australia Health (SA Health) process for RCA as a framework [28]. SA Health is a government-funded entity that maintains public health services such as hospitals and ambulance services across country and metropolitan South Australia and contributes to research and policy development. The purpose of an RCA, as defined by SA Health, is to identify system issues that contribute to an incident and recommend strategies to prevent or minimize the risk of recurrence [28]. The approach focuses on learning from an incident to improve processes or systems used in care delivery. Using an interdisciplinary approach, we followed the first 4 steps of the SA Health RCA process: identification of events, data collection, root cause identification, and recommendation generation [28]. This process was similar to previous aggregate RCAs undertaken in the US, in which hospitalizations from RACS were reviewed, and a previous RCA that examined root causes of falls-related hospitalizations from RACS [30,31]. However, the current study focused specifically on infection-related hospitalizations from an Australian RACS setting. 

### 2.2. Identification of Events

This RCA reviewed 49 infection-related hospitalizations among 41 residents of 6 RACS in South Australia. These infection-related hospitalizations were identified from a previous prospective cohort study that has been described elsewhere [4,32]. Briefly, the previous cohort study followed 383 residents aged ≥ 65 years over 12 months who were recruited from a group of 6 RACS in South Australia. The 383 residents who participated in the previous cohort study were representative of all residents of the 6 RACSs in terms of age, sex, and diagnosed dementia. For the 383 residents participating in the previous cohort study, details of all overnight hospitalizations (e.g., admission date, discharge date, and reason for hospitalization) during follow-up were extracted from RACS records. Hospital transfers that did not result in an overnight stay (e.g., emergency department visits, same day admissions, and outpatient appointments) were not captured. The reasons for all overnight hospitalizations were coded independently by two clinicians according to the World Health Organization International Statistical Classification of Diseases and Related Health Problems, 10th Revision (ICD-10). Infection-related hospitalizations were identified by the following ICD-10 codes: A00-B99, N12, N13.6, N39.0, J09–J18, J22, J44.0, J69.0, L03, and T84.7 [4]. Of the 51 infection-related hospitalizations recorded during the 12 months follow-up, records were available for 49 hospitalizations for inclusion in the present RCA.

### 2.3. Data Collection

The RCA was conducted using a purpose-built tool, developed with input from a multidisciplinary expert panel. The panel consisted of a geriatrician, general medical practitioner (GP), infectious disease physician, 4 pharmacists (infectious diseases pharmacist, geriatrics specialist pharmacist, ambulance service pharmacist, and a community pharmacist who provided collaborative medication reviews in RACS), a consumer representative, and 4 registered nurses. A complete list of panel members appears in the acknowledgments section. The panel included staff from within the RACS provider as well as external experts. Development of this tool was also informed by a review of published literature and the INTERACT Quality Improvement Tool for Review of Acute Care Transfers [30]. The tool was tailored for use in an Australian RACS setting and focused specifically on infection-related hospitalizations. It was designed to capture information about resident characteristics, risk factors, medication use, vaccination status, details of the hospital transfer, actions taken prior to hospital transfer, infection-specific vital signs and symptoms, pathology and other test results, and discharge information provided post-hospitalization. The tool was reviewed for face and content validity by an infectious diseases physician, a pharmacist, and 3 registered nurses.

Using the purpose-built tool described above, a clinical audit nurse extracted data on resident risk factors for infection, changes in resident condition leading up to the hospitalization, and management of risk factors in the RACS. Data were extracted from RACS nursing progress notes, medical records, medication charts, hospital discharge notes, and incident reports for each resident. A pharmacist also reviewed each resident’s medication administration chart to identify medications taken in the month prior to hospitalization that are associated with an increased risk of infection, including oral corticosteroids [33], oxybutynin [34,35], etanercept [36], methotrexate [37], and proton pump inhibitors [38]. Data were collected and managed using the Research Electronic Data Capture (REDCap) web-based tool hosted by Monash University [39]. 

### 2.4. Root Cause Identification

Data extracted for each resident by the clinical audit nurse and pharmacist using the purpose-built tool were reviewed by the research team and an infectious disease physician. The research team were cognizant that clinical staff at the RACS are trained to use a charting by exception approach when documenting information in resident case notes. Factor identification for each case was guided by the SA Health Contributing Factors classification tool, which provides 9 broad factor categories ranging from proximal factors, such as patient assessment, to distal factors, such as facility policies and procedures [28]. For each factor, the research team were asked to apply the 5 why’s technique to ensure the factor represented a root cause rather than a symptom [40,41]. The 5 why’s technique is common approach for conducting RCA in patient safety, and involves the investigator exploring incidents in increasing depth (through continually asking why) until the underlying root cause is identified [31,40,41,42]. An example of using the 5 why’s technique is provided in Table 1. The research team considered events leading up to the hospital transfer and content in the hospital discharge letter, and repeatedly asked “why” when reviewing each key event until the root causes were identified [40,41,42]. The root cause findings were then collated into root cause statements that were grouped according the themes outlined in the SA Health Contributing Factors tool [28]. The root cause statements were tabulated for presentation to the multidisciplinary expert panel.

### 2.5. Recommendation Generation

The same multidisciplinary expert panel that developed the purpose-built tool was reconvened to review all factors that may have contributed to each infection-related hospitalization. For each root cause, panel members were asked to brainstorm potential medication and non-medication interventions that may help to prevent future infection-related hospitalizations. Finally, panel members were asked to review interventions and discuss whether the interventions may be feasible and consider any potential unintended consequences of implementation.

### 2.6. Data Analysis

Descriptive statistics were used to describe the infection, the processes prior to hospitalization, and the resident characteristics at the time of the infection-related hospitalization. The root cause statements and final recommendations of the panel were tabulated according the themes outlined in the SA Health Contributing Factors tool [28].

### 2.7. Ethical Considerations

Ethical approval for this study was obtained from the Monash University Human Research Ethics Committee on November 9, 2017 (application ID 11418). Written informed consent for resident participation in the original cohort study was obtained previously. Panel members provided written informed consent to participate in the RCA. We confirm that the investigations were carried out following the rules of the Declaration of Helsinki of 1975, revised in 2013.

## 3. Results

Among the infection-related hospitalizations reviewed in this study, the median age for residents hospitalized for infection was 86 years (interquartile range 82–92) and 65.3% were female (Table 2). Heart failure (38.8%), chronic obstructive pulmonary disorders (COPD) (34.7%), and diabetes (32.7%) were the most common medical conditions among residents hospitalized for infection. Among residents hospitalized with infection, 12.2% had an indwelling urinary catheter and 20% were taking medications in the month prior to hospitalization that may increase infection risk.

In total, 59.2% of infection-related hospitalizations were for respiratory infections, followed by urinary (28.6%), and skin infections (10.2%) (Table 3). Urinalysis or urinary dipstick testing was undertaken prior to 26.5% of hospital transfers for infection and 17% of residents had blood tests. At the time the infection was suspected and prior to hospital transfer, vital signs were monitored in 81.6% of residents and medications that may increase infection risk were charted in the previous fortnight in 20% of residents. Just over one-third of residents (37.8%) received antimicrobial therapy prior hospital transfer. In four out of five cases (81.6%), the resident’s usual GP, a GP from same practice, or a locum GP had evaluated the resident prior to hospital transfer. Almost three-quarters of hospital transfers for infection occurred on a weekday. Figure 1 shows the time and day of week when each resident was transferred to hospital.

Table 4 lists the factors contributing to infection-related hospitalizations identified in the aggregate RCA. Factors identified include administration of medications that increase the risk of infection. Possible suboptimal selection of empirical antimicrobial therapy and access to medical services including intravenous access, radiology, and pathology were also identified as potential contributors to infection-related hospitalizations. Table 4 also outlines potential strategies to mitigate risk of infection-related hospitalizations as identified by the expert panel. These include strategies such as targeted bundles of care, medication review, antimicrobial stewardship, earlier identification of infection, and models of care that facilitate timely access to medical services.

## 4. Discussion

This was the first Australian aggregate RCA to investigate hospitalizations for infectious diseases from RACS. Factors identified that potentially contributed to infection-related hospitalizations include the use of medications that may increase the risk of infection, selection of empirical antimicrobial therapy, and timely access to subsidized medical, radiology, and pathology services.

Medications that may increase the risk of infection were administered to one in five residents who were hospitalized for infection. It may not be possible to avoid administration of some of these medications, and therefore, prevention and careful monitoring for infection, and early intervention when an infection is present in these “higher risk” residents is important. Potential strategies suggested by the expert panel included medication reviews, implementation of screening tools to identify residents at high risk of infection, embedding flags and decision support tools for high-risk medication use, and education/support for staff.

Respiratory infections and UTIs were identified as the two most common reasons for hospitalization due to infection in our RCA. This is consistent with other studies in the RACS setting [2,5,7,8]. Prevention of respiratory tract infections, in particular pneumonia, is a priority among RACS providers due to associated high rates of morbidity and mortality including hospitalization [43,44]. Prevention strategies include influenza and pneumococcal vaccinations [25]. An infection quality indicator program that includes four indicators pertaining to resident and staff vaccination was recently implemented in public-sector RACS in Victoria, Australia [45]. A recent Cochrane review noted that further research is required to determine whether professional oral care reduces the incidence of pneumonia in comparison to usual oral care [44]. Similarly, prevention of UTIs is important to minimize hospital transfer. A recent systematic review provided a comprehensive list of interventions for prevention of UTIs among residents with and without a urinary catheter [20]. In the present study, only two of the six hospitalizations where an indwelling catheter was present were for UTIs. This may be because the organization involved in this project has implemented a range of strategies to manage residents with urinary catheters including organizational protocols, incontinence nurse reviews, staff training programs, and skills assessments.

An Australian RACS study found that one-third of residents were colonized with at least one antimicrobial-resistant pathogen, including either methicillin-resistant *Staphylococcus aureus*, vancomycin-resistant enterococci, or multidrug-resistant Gram-negative bacilli [12]. The prevalence of multidrug-resistant organisms (MDROs) in RACS is increasing worldwide, with evidence suggesting that some MDROs are more prevalent in RACSs than in acute hospitalized patients [46,47]. A German study reported an average annual cost of €50,306 (USD $56,349) per resident due to antimicrobial-resistant pathogens [48]. Strategies for preventing antimicrobial resistance include monitoring antimicrobial use with a focus on appropriateness [8,49], hand hygiene [13,49], and avoiding unnecessary hospitalization [49]. Infection quality indicators to monitor for three significant organisms (methicillin-resistant *Staphylococcus aureus* and vancomycin-resistant *Enterococcus* and *Clostridium difficile*) have recently been implemented in Victorian public-sector RACS [45].

Selection of suboptimal empirical antimicrobial therapy was identified as a potential factor contributing to infection-related hospitalization. Inappropriate antimicrobial use increases the risk of treatment failure, drug interactions, adverse events, and treatment-related problems such as *Clostridium difficile* infection and contributes to antimicrobial resistance [50]. One of the potential strategies suggested by the expert panel was to optimize antimicrobial use by implementing an interdisciplinary antimicrobial stewardship program. Australian antimicrobial stewardship programs have predominantly focused on the hospital setting, although new Aged Care Quality Standards that apply from July 2019 outline the need for RACS to show evidence of policies and activities to minimize infection-related risks [29]. Since November 2017, multidisciplinary antimicrobial stewardship programs are mandated in all RACS in the US [51]. These programs were introduced to minimize inappropriate antimicrobial use and antimicrobial resistance. An Australian national survey [52] showed that 55.2% of the antimicrobial prescriptions were for residents with no signs and/or symptoms of infection in the week prior to the start date and, of these, only 18.4% met the internationally recognized McGeer et al. [53] infection definitions. Peron et al. found that in the US, 43% of all days of antimicrobial therapy in RACS were unnecessary based on guideline-recommendations [54]. Increased awareness and access to evidence-based resources and guidelines for the management of common infections for health professionals at the RACS was identified by the expert panel as another potential strategy to mitigate risk of hospitalizations due to suboptimal antimicrobial choice. This includes increased on-site and electronic availability to infectious diseases clinical practice guidelines for GPs, locums, other prescribers, and health professionals.

Necessary equipment, appropriately trained staff, and access to external healthcare provider support are required to treat infection within the RACS. These were identified by the expert panel as factors that may contribute to infection-related hospitalizations. Australian RACS provide nursing support rather than acute medical services. Therefore, there is limited capacity for RACS nursing staff to establish intravenous access and administer parenteral antimicrobials [6]. Increasing access to “hospital in the home” or outpatient antimicrobial therapy (OPAT) services to support parenteral antimicrobial administration in RACSs would likely improve resident satisfaction and comfort, minimize length of hospital stay, or avoid the need for hospitalization entirely. Two studies in Australia showed that a “hospital in the home” program could be effective in reducing hospital admissions from RACS residents [55,56]. As part of the RCA, data on the day and time of hospital transfer were recorded because there may be different access to staff and medical services at different times of the day. The availability of staff, equipment, clinical governance, and external clinical support, particularly after hours, have been identified in previous research as barriers to treatment within RACSs [2,18]. This indicates an opportunity that exists to reduce hospital transfers from RACSs by ensuring equipment and expertise are available. One potential solution is presented in a recent evaluation of a “Geriatric Flying Squad” (GFS) model [57]. The team of healthcare providers (the GFS) included a geriatrician, nurse practitioners/nurse practitioner candidates, and clinical nurse consultant who provided a 7-day service. This model involves RACSs referring acutely deteriorating residents to the GP or directly to the GFS if the GP is not contactable. The GFS visit the RACS and provide additional diagnostic and management support not available within the facility. The evaluation indicated that the GFS were able to manage 90.3% of cases within the facility, preventing 578 hospitalizations from RACSs over 18 months. Similarly, a collaborative approach, led by an advanced practice nurse with aged care skills, found that residents receiving this intervention were 41% less likely to be admitted to hospital [58]. Another potential solution may be to better equip primary care practitioners to better manage residents to minimize hospital transfer. This may include providing professional support and education for RACS staff on quality indicators, functional decline, and hospital transfers of residents [59]. Rolland et al. found that this intervention had a significant positive effect on the prevalence of assessment of pressure injury risk, depression, pain, and prevalence of hospital transfers [59]. 

Another factor identified as potentially contributing to hospitalization with infectious diseases was that the resident and/or family member’s wishes regarding hospital transfers may be unknown. Additionally, some advanced care directives may be difficult to interpret and may lack specific information about specific treatments or hospitalizations. In Canada, 21.7% (*n* = 80,413) of residents had “do-not-hospitalize” directives documented between 2009–2010 and 2011–2012, and of these, 7.2% were hospitalized [60]. Among residents who were hospitalized and had a do-not-hospitalize directive, almost half (46.3%) of the hospitalizations were deemed potentially preventable [60]. A potential strategy suggested by the expert panel to mitigate the risk of hospitalization was employing nurse practitioners or training advance care directive “champions” in RACS. This could assist with documentation and interpretation of advanced care directives. A standardized approach to documentation of advanced care directives and specific examples may be important in preventing hospitalizations for infection. 

### Strengths and Limitations

This aggregate RCA recruited residents from six facilities in both metropolitan and rural areas of South Australia. However, the data for the RCA were retrospectively collected from a modest sample of residents, and the residents were recruited from six RACS maintained by one aged care provider organization, and therefore, generalizability may be limited. However, the 383 residents included in the original cohort study from which this aggregate RCA was derived were representative of all residents of the 6 RACS in terms of age, sex, and diagnosed dementia. Although the sample size is small compared to epidemiological studies, our study is based on a comprehensive and in-depth review of nursing progress notes, medical records, medication charts, hospital summaries, and incident reports for each of the participating residents. These data were reviewed independently by the research nurse, an infectious diseases physician, and an infectious diseases pharmacist. Additionally, the expert panel was composed of members internal and external to the RACS provider ensuring that reviews were well informed and independent. A lack of independence has previously been a criticism of RCA [61]. The purpose-specific data collection tool was based on the SA Health RCA tool. The tool was developed by the expert panel ensuring that all relevant information was captured to inform the RCA. While single incident analysis may lead to prioritizing actions and resources to a rare event, an aggregate RCA identifies recurring events allowing for consideration of system and human factors contributing to hospitalizations [61]. A limitation of our approach is that by only assessing infection-related hospitalizations, we were unable to ascertain whether factors perceived to contribute to infection-related hospitalizations were different to those which may contribute to hospitalizations for other health conditions. However, this is an important issue to examine because 25% of hospitalizations from RACS are for infection [4].

## 5. Conclusions

This aggregate RCA identified medication and non-medication opportunities that exist to prevent infection-related hospitalizations through targeted medication review, antimicrobial stewardship, earlier identification of infection, and models of care that facilitate timely and extended access to medical services. RACS provider organizations, clinicians, policy makers, and other stakeholders can use these findings to review current strategies in place and inform next steps to limit infections and associated hospital transfers from RACS. Future studies could explore factors associated with successful implementation and associated outcomes for residents and other stakeholders.

## Figures and Tables

**Figure 1 ijerph-17-03282-f001:**
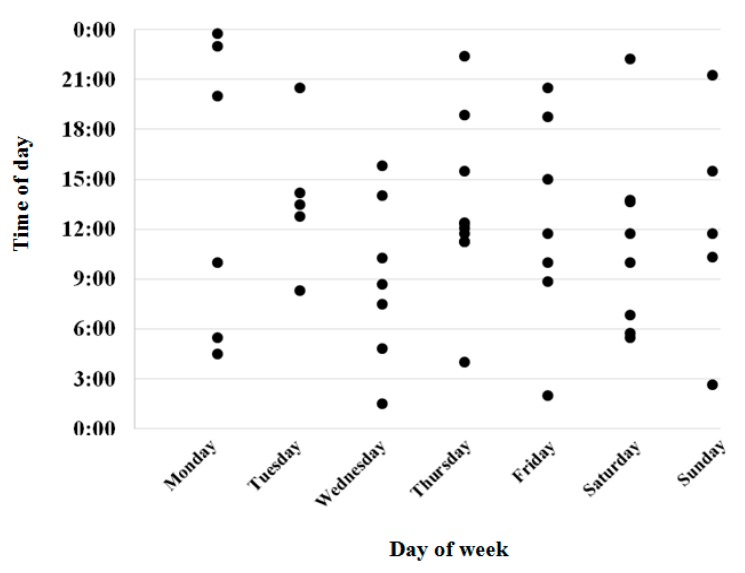
Time and day of hospital transfer among residents hospitalized for infection (*n* = 49).

**Table 1 ijerph-17-03282-t001:** Example of using the 5 why’s technique.

The resident showed signs of deterioration after a recent hospitalization for infection—why? Because the resident did not receive an antibiotic—why? Because the antibiotic was not commenced on return to the residential aged care service as recommended in the hospital discharge letter—why? Because the resident’s usual general medical practitioner did not initiate the antibiotic on return to the residential aged care service—why? Because the actions to take post-discharge were not clearly outlined in the hospital discharge letter

**Table 2 ijerph-17-03282-t002:** Resident characteristics at the time of the infection-related hospitalization (*n* = 49 hospitalizations).

Characteristic	*N* (%) or Median (Interquartile Range)
Age (Years)	86 (82–92)
Female	32 (65.3)
Medical conditions Dementia Diabetes Asthma Chronic obstructive pulmonary disease Ischemic heart disease Heart failure Prior stroke	10 (20.4) 16 (32.7) 8 (16.3) 17 (34.7) 16 (32.7) 19 (38.8) 7 (14.3)
Current smoker	2 (4.1)
Indwelling catheter	6 (12.2)
History of infection in the previous 6 months	15 (30.6)
Advance care directive in place prior to hospitalization	34 (69.4)
Medication use ^a^ Polypharmacy (≥9 regular medications) ^b^ Charted regular medications that may increase infection risk ^b^ Influenza vaccination prescribed by GP on the RACS medication chart and documented as administered ^c^	32 (71.1) 9 (20.0) 11 (24.4)

Abbreviations: GP, general medical practitioner; RACS, residential aged care service. ^a^ Medication chart for the 2 weeks prior to hospitalization was not available on four occasions. ^b^ Assessed in the 2 weeks prior to the hospitalization. ^c^ Assessed in the 12 months prior to hospitalization. It is noted that influenza vaccinations may sometimes be recorded by GPs as administered in the progress notes only.

**Table 3 ijerph-17-03282-t003:** Characteristics of the infection and resulting infection-related hospital transfer.

Characteristic	*N* (%) (*n* = 49)
Infection type Respiratory infection Pneumonia Exacerbation of chronic obstructive pulmonary disease Urinary infection Urinary tract infection Urosepsis Hospitalizations for urinary infection where an indwelling catheter was present Skin infection Cellulitis Other	29 (59.2) 12 (24.5) 5 (10.2) 14 (28.6) 7 (14.3) 6 (12.2) 2 (4.1) 5 (10.2) 3 (6.1) 4 (8.2)
New or worsening signs or symptoms in the 2 weeks prior to hospital transfer Feeling unwell Altered mental status or changes in behavior Malaise, lethargy, drowsiness, or refusal to get out of bed Functional decline Fall New or worsening pain Fever, chills, or rigors Decreased oral intake Nausea or vomiting New/increasing abdominal pain or diarrhea	15 (30.6) 7 (14.3) 13 (26.5) 9 (18.4) 7 (14.3) 17 (34.7) 7 (14.3) 10 (20.4) 13 (26.5) 8 (17.0)
Testing undertaken within the RACS in the 2 weeks prior to hospital transfer ^a^ Blood test Urinary dipstick or urinalysis Other Radiology No testing undertaken	8 (17.0) 13 (26.5) 3 (6.4) 0 (0.0) 29 (61.7)
Interventions undertaken within the RACS from the time the condition was first suspected until hospital transfer Monitor vital signs New or change in medication(s) Oxygen Physiotherapy review/treatment Other None required	40 (81.6) 27 (55.1) 22 (44.9) 3 (6.1) 5 (10.2) 3 (6.1)
External provider evaluation of the resident Usual GP or GP from same practice Locum GP Nurse Practitioner Extended care paramedic Resident’s condition discussed with GP or locum via telephone Nil documented	27 (55.1) 13 (26.5) 1 (2.0) 5 (10.2) 5 (10.2) 6 (12.2)
Antimicrobial use in the 2 weeks prior to hospital transfer ^b^ Penicillin Cephalosporin Macrolide Trimethoprim or nitrofurantoin Oseltamivir	17 (37.8) 9 (20.0) 5 (11.1) 5 (11.1) 4 (8.9) 2 (4.4)
Person authorizing hospital transfer Usual GP or GP from same practice Locum GP Nurse practitioner Registered nurse Resident or family member Extended care paramedic	13 (26.5) 8 (16.3) 2 (4.1) 19 (38.8) 4 (8.2) 3 (6.1)
Day of hospital transfer Weekday (Monday–Friday) Weekend (Saturday–Sunday)	36 (73.5) 13 (26.5)
Time of hospital transfer Between 07:00 and 14:59 Between 15:00 and 22:59 Between 23:00 and 06:59	24 (49.0) 13 (26.5) 12 (24.5)

Abbreviations: GP, general medical practitioner; RACS, residential aged care service. ^a^ Information was available for *n* = 47 events. ^b^ Medication administration charts for the 2 weeks prior to hospitalization were available for *n* = 45 events.

**Table 4 ijerph-17-03282-t004:** Factors contributing to infection-related hospitalizations identified through the root cause analysis and potential strategies to mitigate the risk of hospitalization that were identified by panel members.

Domain	Factors Contributing to Infection-Related Hospitalizations Identified through the Root Cause Analysis	Potential Strategies to Mitigate Risk of Infection-Related Hospitalizations
Resident assessment	Administration of medications that increase the risk of infection (e.g., corticosteroids) or contribute to urinary retention (e.g., medications with anticholinergic properties) Possible suboptimal management of adrenal insufficiency during acute infection Possible suboptimal selection of empirical antimicrobial therapy	Consider implementation of a screening tool to identify residents who are at high risk of infection Increase awareness and access to evidence-based resources and guidelines for management of common infections and increase on-site and electronic availability (e.g., Therapeutic Guidelines) Embed flags and decision support tools relating to identification of medication use that may increase infection risk, identify residents at risk of adrenal insufficiency during acute infection, and support optimal empirical antimicrobial selection into electronic RACS medication management systems, where available in the RACS Increase awareness and access to tools to facilitate regular review of skin care in residents at high risk of skin infections (e.g., those with diabetes or using topical corticosteroids for extended periods) Increase awareness and access to tools to monitor fluid balance Implement a subsidized RACS antimicrobial stewardship program that is adequately resourced to bring together GPs, facility staff, pharmacists, and external infectious disease physician expertise Clinical pharmacist or nurse employed within the RACS as part of a subsidized program to undertake antimicrobial stewardship
Staff training and resident factors	Earlier identification and response to signs and symptoms of confusion, delirium, infection, and sepsis Earlier recognition and response to signs and symptoms of reduced oral intake and dehydration as early signs of infection Possible deficits in knowledge and practices relating to specimen collection Possible inconsistent documentation of observations where indicated (e.g., documented in the progress notes and/or observation chart) Inhaler technique may not be regularly checked or corrected by a health professional	Implement a structured checklist and training package to support clinical staff to identify signs and symptoms of dehydration, infection, and sepsis Develop and implement a clinical pathway to assist staff to respond to suspected infections Implement a subsidized “diagnostic stewardship” program that is adequately resourced to engage GPs and clinical RACS staff Increase awareness and access to existing chronic obstructive pulmonary disease and asthma action plans Involve pharmacists in the review of inhaler technique, training for staff/residents and provision of chronic obstructive pulmonary disease and asthma action plans Increased access to “hospital in the home” or similar external service to support parenteral rehydration in residents with limited oral intake and dehydration
Equipment and work environment	Problems with timely access to subsidized medical, radiology, and pathology services RACS clinical staff unable to establish intravenous access and administer parenteral antimicrobials at the RACS	Increase access to mobile or on-site pathology and radiology services that are subsidized for residents Utilize telehealth services to facilitate review and inform the decision to initiate a hospital transfer Develop and implement subsidized models of care that support proactive on-site multidisciplinary care from GPs and geriatricians Increased access to “hospital in the home” or external OPAT services to support parenteral antimicrobial administration in RACS to support hospital avoidance or early discharge Models that support input from infectious diseases physicians during infectious disease outbreaks that may occur within RACS
Information, policies, and procedures	The resident and/or family member’s wishes regarding hospital transfers may be unknown Influenza vaccinations were not always prescribed and/or there may be difficulty in determining current vaccination status. Pneumococcal vaccination status was difficult to determine as residents may have been immunized many years prior to admission to the RACS but documentation regarding administration may not have been received from the previous GP and/or the resident or family may not be able to provide vaccination history when the resident first enters the RACS	A specific procedure to support documentation of resident’s wishes (e.g., advance care directives) in a clear and consistent manner to inform decision-making regarding a hospital transfer for infection Nurse practitioners or advance care directive “champions” within a RACS could assist with documentation of advanced care directives (implemented since completion of study) Support health professionals to reference existing and emerging tools (e.g., electronic health records such as Australia’s My Health Record or immunization registers such as the Australian Immunization Register) to record vaccines given to residents Embed flags to highlight future immunization dates into electronic RACS medication management systems where available and in use Robust procedures in place to ensure immunizations are administered and this is documented for RACS staff to view
Communication and coordination	Challenges with timely communication between health professionals and staff at RACS when changes occur in resident behavior, cognition, physical status, and medications Delays in reviewing pathology test results received post-initiation of empirical antimicrobial therapy Suboptimal communication of results of pathology tests undertaken in hospital and ongoing antimicrobial therapy plan after hospital discharge	Facilitate timely communication of changes in resident behavior, cognition, and medication use to all persons involved in the resident’s care Implement a standardized format for transfer of information, e.g., ISBAR Facilitate timely access to review of empirical therapy through mechanisms such as antimicrobial stewardship programs

Abbreviations: GP, general medical practitioner; ISBAR, Introduction, Situation, Background, Assessment, Recommendation; OPAT, outpatient antimicrobial therapy; RACS, residential aged care service.

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
