# Peer review of "Root Cause Analysis to Identify Medication and Non-Medication Strategies to Prevent Infection-Related Hospitalizations from Australian Residential Aged Care Services"

_ijerph, 2020, doi:10.3390/ijerph17093282_

Round 1

Reviewer 1 Report

The central topic of the work is hospitalizations for infections by patients from residential aged care services and the strategy to reduce them, a very important topic from a public health perspective.
In the introduction we start talking about Coronavirus disease even if it's not a topic covered in the work maybe it's superfluous to include that phrase.
The work is very interesting because it deals with the clinical risk assessment applied in 6 RACS. The topic is very relevant and the study is conducted in a rigorous and quality approach.
There is a typo: line 111 is missing the square bracket before the numbers of the bibliography.
The bibliography is updated.
The authors in the final discussion identify what further research might be needed in this field.

Author Response

Reviewer comment 1. The central topic of the work is hospitalizations for infections by patients from residential aged care services and the strategy to reduce them, a very important topic from a public health perspective. The work is very interesting because it deals with the clinical risk assessment applied in 6 RACS. The topic is very relevant and the study is conducted in a rigorous and quality approach. The bibliography is updated. The authors in the final discussion identify what further research might be needed in this field.

Response: We thank the reviewer for this positive feedback.

Reviewer comment 2. In the introduction we start talking about Coronavirus disease even if it's not a topic covered in the work maybe it's superfluous to include that phrase.

Response: We removed the text that mentioned Coronavirus in response to comments from Reviewers 1 and 2.

Reviewer comment 3. There is a typo: line 111 is missing the square bracket before the numbers of the bibliography.

Response: We corrected this typographical error in the revised version.

Reviewer 2 Report

The paper discusses the aggregate root cause analysis of infection related hospitalization from residential aged care centers. The paper analyzed systematically the problem and discussed precise way. However, the author mentioned the Coronavirus disease in introduction section but not discussed further in discussion section.

Author Response

Reviewer comment 1. The paper discusses the aggregate root cause analysis of infection related hospitalization from residential aged care centers. The paper analyzed systematically the problem and discussed precise way. However, the author mentioned the Coronavirus disease in introduction section but not discussed further in discussion section.

Response: We thank the reviewer for this positive feedback. We removed the text that mentioned Coronavirus in response to comments from Reviewers 1 and 2.

Reviewer 3 Report

Title:

The authors may need to consider including the area, Australia, in their title.

Abstract:

Include a brief description of “residential aged care services” and “Root cause identification” 

What are the aim and significance of the study? Also, the authors need to describe research methods.

Keywords: Is there any difference between residential aged care and a nursing home? I think the authors can take nursing home out from the keywords.

Introduction:

The authors may want to provide a justification for choosing Australia as a research area.

Line 97-98: Are you including residential aged care elderly in this study? If so, the authors may want to include that at the end of the sentence. Also, the authors may want to state the significance of this study (e.g., challenge or existing knowledge).

Is there any framework or theory applied in this study?

Materials and methods:

Lines 106-107: this should appear earlier in the introduction. Also, I encourage the authors to pick the most suitable term for their research and use it consistently.

Line 126: more description of “a purpose-built tool” is needed

Line 153: what is the 5 why’s technique? Please describe and give examples.

The data analysis section is lacking (e.g., how did you validate the data analysis?) and the root cause analysis is unclear.

Results:

Not all information in Table 2 seemed to be necessary. For example, what is the rationale of including the day or time of hospital transfer? Only include information relevant to the authors’ research question. Same comment for figure 1. What does figure 1 tell regarding the authors’ research question?

Discussion:

Lines 234-235: this information should appear in the introduction section.

The authors may want to include suggestions for future studies.

Author Response

Reviewer comment 1. Title: The authors may need to consider including the area, Australia, in their title.

Response: We made the suggested change and the revised title is: “Root Cause Analysis to Identify Medication and Non-medication Strategies to Prevent Infection-related Hospitalizations from Australian Residential Aged Care Services”.

Reviewer comment 2. Abstract: Include a brief description of “residential aged care services” and “Root cause identification”.

Response: We briefly describe the term RACS in the abstract of the revised paper, and we edited the sentence describing root cause identification.

Reviewer comment 3. Abstract: What are the aim and significance of the study? Also, the authors need to describe research methods.

Response: We revised the abstract to present a structured aim statement and presented further justification for the significance of the study. The revised text states:

“Preventing infections and early and effective management are important to avoid unnecessary hospital transfers, particularly in the Australian setting where new quality standards require RACS to minimize infection-related risks. The objective of this study was to examine root causes of infection-related hospitalizations from RACS and identify strategies to limit infections and avoid unnecessary hospitalizations.”

We revised the methods section to provide further clarity:

“An aggregate root cause analysis (RCA) was undertaken using a structured local framework. A clinical nurse auditor and clinical pharmacist undertook a comprehensive review of 49 consecutive infection-related hospitalizations from six RACS. Data were collected from nursing progress notes, medical records, medication charts, hospital summaries and incident reports using a purpose-built collection tool. The research team then utilized a structured classification system to guide the identification of root causes of hospital transfers. A multidisciplinary clinical panel assessed the root causes and formulated strategies to limit infections and hospitalizations.”

Reviewer comment 4. Keywords: Is there any difference between residential aged care and a nursing home? I think the authors can take nursing home out from the keywords.

Response: We removed the key word nursing home as suggested.

Reviewer comment 5. Introduction: The authors may want to provide a justification for choosing Australia as a research area.

Response: We added the following text to further explain the reason for this Australian study:

“In Australia, the formal, subsidized interdisciplinary antimicrobial stewardship programs that exist within the hospital setting and not routinely available in Australian RACS. However, new national Aged Care Quality Standards that apply from July 2019 outline the need for RACS provider organizations to implement antimicrobial stewardship policies and activities [29]. One quarter of all hospitalizations from South Australian RACS are for infections [4].”

In the introduction, we also highlight this novel aggregate RCA approach used in the present study has not been used to comprehensively investigate infection-related hospital transfers in Australia previously.

Reviewer comment 6. Line 97-98: Are you including residential aged care elderly in this study? If so, the authors may want to include that at the end of the sentence. Also, the authors may want to state the significance of this study (e.g., challenge or existing knowledge).

Response: We now mention at the end of the aim statement that the study included residents of aged care services (page 3, line 109-110). In the methods section we have described that the study included residents aged 65 years and older (page 3, section 2.2). We have provided additional information to this paragraph in response to Reviewer 3 comment 5 above to highlight the importance of the study locally.

Reviewer comment 7. Is there any framework or theory applied in this study?

Response: We utilized the SA Health root cause analysis framework to guide this study. We have added the following information to Section 2.1 of the methods to further explain this approach:

“This study was an aggregate RCA that utilized the South Australia Health (SA Health) process for RCA as a framework [28]. SA Health is a government-funded entity that maintains public health services such as hospitals and ambulance services across country and metropolitan South Australia and contributes to research and policy development. The purpose of an RCA, as defined by SA Health, is to identify system issues that contribute to an incident and recommend strategies to prevent or minimize the risk of recurrence [28]. The approach focuses on learning from an incident to improve processes or systems used in care delivery. Using an interdisciplinary approach, we followed the first 4 steps of the SA Health RCA process…”

Reviewer comment 8. Materials and methods: Lines 106-107: this should appear earlier in the introduction. Also, I encourage the authors to pick the most suitable term for their research and use it consistently.

Response: We moved the definition of an RACS to the introduction as suggested by the reviewer. We have reviewed the text to ensure the term ‘RACS’ is used consistently throughout the manuscript.

Reviewer comment 9. Line 126: more description of “a purpose-built tool” is needed

Response: We have described that development of the tool was informed by a literature review and the INTERACT Quality Improvement Tool for Review of Acute Care Transfers. We added the following text to describe the content of the purpose-built tool:

“It was designed to capture information about resident characteristics, risk factors, medication use, vaccination status, details of the hospital transfer, actions taken prior to hospital transfer, infection-specific vital signs and symptoms, pathology and other test results, and discharge information provided post-hospitalization.”

Reviewer comment 10. Line 153: what is the 5 why’s technique? Please describe and give examples.

Response: The 5 why’s technique is a standard technique employed in root cause analyses. We added the following text, several references and a new Box 1 to provide further information on the 5 why’s process used for this RCA:

“For each factor the research team were asked to apply the 5 why’s technique to ensure the factor represented a root cause rather than a symptom [40, 41]. The 5 why’s technique is common approach for conducting RCA in patient safety, and involves the investigator exploring incidents in increasing depth (through continually asking why) until the underlying root cause is identified [31, 40–42]. An example of using the 5 why’s technique for this RCA is provided in Box 1. The research team considered events leading up to the hospital transfer and content in the hospital discharge letter, and repeatedly asked ‘why’ when reviewing each key event until the root causes were identified [40–42]. The root cause findings were then collated into root cause statements that were grouped according the themes outlined in the SA Health Contributing Factors tool [28].”

Reviewer comment 11. The data analysis section is lacking (e.g., how did you validate the data analysis?) and the root cause analysis is unclear.

Response: We added new section 2.6 to clarify the data analysis:

“2.6. Data analysis

Descriptive statistics were used to describe the infection, the processes prior to hospitalization and the resident characteristics at the time of the infection-related hospitalization. The root cause statements and final recommendations of the panel were tabulated according the themes outlined in the SA Health Contributing Factors tool [28].”

We added the following text to the manuscript to clarify the root cause analysis:

“The root cause findings were then collated into root cause statements that were grouped according the themes outlined in the SA Health Contributing Factors tool [28]. The root cause statements were tabulated for presentation to the multidisciplinary expert panel.”

Reviewer comment 12. Results: Not all information in Table 2 seemed to be necessary. For example, what is the rationale of including the day or time of hospital transfer? Only include information relevant to the authors’ research question. Same comment for figure 1. What does figure 1 tell regarding the authors’ research question?

Response: The rationale for including the day and time of hospital transfer was that there may be different access to staff and medical services at different times of the day. Staffing issues and access to medical services are frequently cited in the literature and factors involved in the decision to transfer a resident to hospital. Even in the absence of a clear association between the day and time with hospital transfer, these data are important for workforce planning.

We added the following information to the discussion section (lines 313-315):

“As part of the RCA, data on the day and time of hospital transfer were recorded because there may be different access to staff and medical services at different times of the day.”

Reviewer comment 13. Discussion: Lines 234-235: this information should appear in the introduction section.

Response: We moved this sentence to the introduction section as suggested by the reviewer.

Reviewer comment 14. The authors may want to include suggestions for future studies.

Response: We added the following text to the conclusion to highlight next steps:

“RACS provider organizations, clinicians, policy makers and other stakeholders can use these findings to review current strategies in place and inform next steps to limit infections and associated hospital transfers from RACS. Future studies could explore factors associated with successful implementation and associated outcomes for residents and other stakeholders.”

Round 2

Reviewer 3 Report

Thank you for addressing all of my comments. The revised manuscript has been much improved. Great job!